

# High Altitude Aerosol Chemical Characterization and Source Identification: Insights from the CALISHTO Campaign

Olga Zografou[1,2], Maria Gini[1], Prodromos Fetfatzis[1], Konstantinos Granakis[1], Romanos Foskinis[1,3,8], Manousos Ioannis Manousakas[1,4], Fotios Tsopelas[2], Evangelia Diapouli[1], Eleni Dovrou[5,6], Christina N. Vasilakopoulou[5,6], Alexandros Papayannis[3], Spyros N. Pandis[5,6], Athanasios Nenes[7,8], and Konstantinos Eleftheriadis[1]

[1]Environmental Radioactivity & Aerosol Tech. for Atmospheric & Climate Impacts, INRaSTES, National Centre of Scientific Research "Demokritos", Ag. Paraskevi, 15310, Greece
[2]Laboratory of Inorganic and Analytical Chemistry, School of Chemical Engineering, National Technical University of Athens, Athens, Greece
[3]Laser Remote Sensing Unit, Physics Department, School of Applied Mathematics and Physical Sciences, National and Technical University of Athens, 15780 Zografou, Greece
[4]Laboratory of Atmospheric Chemistry, Paul Scherrer Institute, CH-5232, Villigen PSI, Switzerland
[5]Institute of Chemical Engineering Sciences, ICE-HT, Patras, 26500, Greece
[6]Department of Chemical Engineering, University of Patras, Patras, 26504, Greece
[7]Institute for Chemical Engineering Sciences, Foundation for Research and Technology, Patras, Greece
[8]School of Architecture, Civil and Environmental Engineering, École Polytechnique Fédérale de Lausanne, Lausanne, Switzerland

*Correspondence to*: Konstantinos Eleftheriadis (elefther@ipta.demokritos.gr) and Olga Zografou (o.zografou@ipta.demokritos.gr)

**Abstract.** The Cloud-AerosoL InteractionS in the Helmos background TropOsphere (CALISHTO) campaign took place in autumn 2021 at the NCSR Demokritos background high-altitude Helmos Hellenic Atmospheric Aerosol & Climate Change station (HAC)[2] to study the interactions between aerosols and clouds. The current study presents the chemical characterization of the Non-Refractory (NR) $PM_1$ aerosol fraction using a Time-of-Flight Aerosol Chemical Speciation Monitor (ToF-ACSM). A comparative offline aerosol filter analysis by a High-Resolution Time-of-Flight Aerosol Mass Spectrometry (HR-ToF-AMS) showed consistent results regarding the species determined. Source apportionment applied on both datasets (ACSM-ToF and offline AMS analysis on filter extracts) yielded the same factors for the organic aerosol (one primary and two secondary factors). Additionally, the Positive Matrix Factorization (PMF) model was applied on the total $PM_1$ fraction by the ToF-ACSM (including both organic and inorganic ions). Five different types were identified, including a primary organic factor, ammonium nitrate, ammonium sulphate, and two secondary organic aerosols; one more and one less oxidized. The prevailing atmospheric conditions at the station, i.e. cloud presence, influence from emissions from the Planetary Boundary Layer (PBL) and air mass origin were also incorporated in the study. The segregation between PBL and Free Troposphere (FT) conditions was made by combining data from remote sensing and in-situ measurement techniques. The types of air masses arriving at the site were grouped as continental, marine, dust and marine-dust based on back trajectories data. Significant temporal variability in the aerosol characteristics was observed throughout the campaign; in September, air masses from within the PBL were



sampled most of the time, resulting in much higher mass concentrations compared to October and November when concentrations were reduced by a factor of 5. Both in-cloud and FT measurement periods resulted in much lower concentration levels, while similar composition was observed in PBL and FT conditions. We take advantage of using a recently developed "virtual filtering" technique to separate interstitial from activated aerosol sampled from a $PM_{10}$ inlet during cloudy periods.

This allows the determination of the chemical composition of the interstitial aerosol during in-cloud periods. Ammonium sulphate, the dominant PMF factor in all conditions, contributed more when air masses were arriving at (HAC)[2] during Dust events, while higher secondary organic aerosol contribution was observed when air masses arrived from continental Europe.

## 1 Introduction

Atmospheric aerosols exhibit a large diversity regarding their sources, size distribution, chemical composition, and lifetime
across the globe. Clouds play a crucial role on climate, the hydrologic cycle, and on the lifecycle of gaseous species and particulate, owing to their contribution to deposition pathways, and offering the medium for aqueous phase reactions (Seinfeld and Pandis, 2006). Atmospheric aerosol, serving as Cloud Condensation Nuclei (CCN) and Ice Nuclei (IN) provide the seeds upon which droplets and ice crystals can form; modulations of aerosol abundance and type from anthropogenic or natural sources therefore can have important impacts on climate and the hydrological cycle. It is now established that anthropogenic
aerosol impacts on clouds and climate have led to its cooling, but its large uncertainty also impedes the ability to constrain the climate sensitivity to greenhouse gas warming (IPCC, 2021). Clouds are impacted by aerosol modulations, but clouds also affect aerosols, as cloud microphysical processes (e.g., coagulation of droplets and ice crystals, collection of interstitial particles by droplets and ice crystals, in-cloud chemistry) lead to changes in the aerosol size distribution and chemical composition after the evaporation of cloud droplets, differing from the precursor aerosol particles (Roth et al., 2016). CCN
usually originate from the accumulation mode and activate into cloud droplets that grow, in the absence of drizzle or precipitation, to sizes that range between 5 and 20 μm radius (e.g., Seinfeld and Pandis, 2016). Interstitial particles are the smaller aerosol particles that remain inactivated. Coagulation takes place between cloud droplets and the interstitial particles, resulting in the so-called in-cloud scavenging of particles. Observational constraints of such in-cloud processes are key for constraining models of aerosol-cloud interactions. The study of aerosol-cloud interactions at the cloud microphysical scale
requires relevant in-situ measurements, which can be carried out using airborne platforms (tethered balloons, aircraft, UAV) – observations of orographic clouds with ground-based infrastructure also allow for the direct characterization of aerosol and cloud microphysical processes over extended periods of time. In such studies, a key issue is to understand the origin of aerosol upon which droplets and ice crystals form on.

Mountainous atmospheric measurement stations are often influenced by Planetary Boundary Layer (PBL) aerosol, either
because the station resides within the PBL at certain periods of the day/season or by aerosol convection from the PBL up into the Free Troposphere (FT). The PBL is the lowest part of the atmosphere and is characterized by turbulence that tends to mix the aerosol within it (Stull, 2016). The part of the atmosphere between 2 and 11 km from the ground is considered as FT,



containing air unperturbed by turbulence (Stull, 2016). In general, within the PBL, solar heating of the ground surface during daytime leads to intense mixing and growth of the PBL height, while cooling during nighttime leads to a contraction of the PBL. In the case of mountainous regions, katabatic winds consist another source of mixing, additional to the expansion and contraction of the PBL height. This diurnal cycle in the PBL's height has a great influence in dispersion and vertical transport of pollutants in addition to horizontal wind. Specifically for Helmos Mt., Foskinis et al (in review) studied the PBL height (PBLH) for a 7-month period and showed that starting from September there is a pronounced diurnal trend of PBLH which exceeds the station's height at noon. During November the diurnal variability is rather flat and the station appears to be in the entrainment zone, while December to February the PBL is mostly lower than the station's altitude. Starting March, a diurnal variability appears again and more often the PBLH exceeds that of the station. Removal of aerosols is slower in the FT than in the PBL, since cloud presence is more common in the PBL (therefore lower wet removal in the FT) and turbulent mixing is more important in the PBL, resulting in higher dry deposition. FT aerosols have generally longer lifetimes and more significant impact contribution to the direct effect on climate (e.g., Pandis et al., 1992), as opposed to aerosols within the PBL that strongly influence low level clouds and hence the indirect climate effect (IPCC, 2021).

Fröhlich et al. (2015) introduced the Time-of-Flight Aerosol Chemical Speciation Monitor (ToF-ACSM) over a 14-month measurement campaign in the Jungfraujoch station and showed great influence from anthropogenic activities despite its high altitude (3580 m a.s.l.). Other studies reporting Particulate Matter (PM) chemical composition by mass spectrometry from high altitude stations include Ripoll et al. (2015) for Montsec in Spain (1570 m a.s.l.), Farah et al. (2021) for Puy-de-Dôme – PUY station (1465 m a.s.l.), Rinaldi et al. (2015) for Mt. Cimone (2165 m a.s.l.), Mukherjee et al. (2018) and Singla et al. (2019) for HACPL in India (1378 m a.s.l.), and others. Great variability is observed concerning the mass loading in high-altitude stations, as well as the chemical composition of $PM_1$ (Zhou et al., 2018), with respect not only to the height of each station and the season studied, but also to the impact of PBL emissions on the measurements (Collaud Coen et al., 2018). However, none of these studies discussed aerosol-cloud interactions with respect to chemical composition.

The Positive Matrix Factorization (PMF) model is most commonly applied on the organic fraction of real-time mass spectrometry datasets to identify prevailing sources of OA. In JFJ, Fröhlich et al (2015) retrieved a Hydrocarbon-related OA factor (HOA) and a local primary OA factor for all seasons, while one or two Oxygenated OA (OOAs) factors were retrieved depending on the season. At PUY, Farah et al (2021) identified one HOA, and one OOA factor in all seasons, and one biomass-burning related factor (BBOA) only in spring. The same factors were identified in winter at MSC (Ripoll et al., 2015), while HOA with two OOAs were retrieved for the summer period at this site. Rinaldi et al (2015) found only three OOAs and no influence from primary emissions. One HOA, one BBOA and one to two OOA factors, depending on the season, were also identified in HACPL station (Mukherjee et al., 2018). While in the same dataset, two factors were added when combined PMF analysis took place; one nitrate-OA and one sulphate-OA. Zhou et al (2018) combined organic and inorganic ions for PMF analysis, and presented a 3-factor solution consisting of two OOAs, one of which contained sulphate ions, and one sulphate-dominated OOA factor.



The Cloud-AerosoL InteractionS in the Helmos background TropOsphere (CALISHTO) campaign took place in autumn 2021 at the NCSR Demokritos background high-altitude Helmos Hellenic Atmospheric Aerosol & Climate Change (HAC)[2] station to study the interactions between aerosols and clouds. Here we focus on deepening our knowledge on the effect of aerosol-cloud interactions to the chemical composition of the background atmosphere, to characterize the chemical fingerprint and

sources of the air masses at a high-altitude station based on their origin and with respect to PBLH. Finally, we aimed to establish trustworthy metrics for resolving the origin from within or above the PBL using observations at the (HAC)[2] station that can be applied in the long-term in absence of remote sensing instrumentation. To differentiate between activated and interstitial particles, key to our analysis, we followed the "virtual filtering technique" proposed by Foskinis et al (in review), in which a sensitivity analysis took place on the cut-off size of the effective diameter of the cloud droplets, as provided by the

cloud probe, to determine the size up to which both interstitial and activated particles are being sampled. Results of our analysis are also compared with those derived from offline analysis of filter extracts that is aerosolized and introduced into an Aerodyne Aerosol Mass Spectrometer (AMS; Vasilakopoulou et al., 2023).

## 2 Experimental

### 2.1 Measurement site

The (HAC)[2] station (Latitude 37.9842 N, Longitude 22.1969 E) is located at mountain top of Helmos (or Aroania) mountain, situated in North Peloponnese, Greece (Figure S1). Mt. Helmos is the only high-altitude station in the eastern Mediterranean region. At an altitude of 2314 m a.s.l., the location of the station allows the study of interactions between aerosols and clouds, as it is often in-cloud, especially in the fall and winter periods (Foskinis et al., in review). It is the station with the lowest ABL-TopoIndex in Europe, according to Collaud Coen et al (2018), meaning that it has, compared to other mountaintop sites, fewer

PBL influences and is therefore favorable for characterizing FT aerosols. Nevertheless, PBL influences do exist and can be very important depending on season and time-of-day (Foskinis et al., in review) – which if well-constrained provides an additional advantage for studying aerosol-cloud interactions from aerosol types that are emitted nearby or regionally but are aged in the boundary layer (e.g., bioaerosols from the nearby forest or regional biomass burning; Gao et al., in review). Additionally, (HAC)[2] is situated in a location where air masses from different origins arrive, including continental, Saharan,

and long-range biomass burning. This facilitates the study of ambient PM with markedly different properties. It is also a contributing station within the Global Atmosphere Watch (GAW) programme, as well as submitting data to ACTRIS (actris.eu) under the acronym HAC (Rose et al., 2021) and part of several infrastructures, including the PANhellenic infrastructure for Atmospheric Composition and climatE chAnge (PANACEA).

### 2.2 Instrumentation

During the CALISHTO campaign, a large suite of instrumentation was deployed that included in-situ and remote sensing instruments (https://calishto.panacea-ri.gr/) at (HAC)[2] as well as at the temporary site of Vathia Lakka (VL) (1850 m a.s.l.)



and the nearby Kalavryta Ski Resort. Central to this study are the aerosol chemical composition data collected from the ToF-ACSM (Aerodyne Research Inc., Billerica, MA, USA) deployed at (HAC)[2], which provides information on the aerosol chemical composition at high temporal resolution. The ToF-ACSM carries many similarities to the Aerosol Mass Spectrometer (AMS), and its operation principles are described in detail by Fröhlich et al. (2015). In summary, a $PM_{2.5}$ cut-off inlet equipped with a Nafion drier is installed, and Non-Refractory Species (NRS; organics, sulphate, nitrate, ammonium and chloride) of $PM_1$ are detected, after their vaporization and ionization, through a detector. The Relative Ionization Efficiencies (RIEs) for organics, $NO_3^-$ and $Cl^-$ were 1.4, 1.1 and 1.3, respectively (Fröhlich et al., 2015). After performing calibrations at the (HAC)[2] station, the RIEs for sulphate and ammonium were found to be 1.19 and 3.11, respectively. To maintain the inlet mass flow rates at relevant levels compared to those in low altitude operation, a different orifice with a diameter of 120 μm was placed instead of the regular 100 μm orifice (Fröhlich et al., 2015). According to Middlebrook et al (2012), a Collection Efficiency (CE) needs to be applied to correct for particle losses during collection, and depends on the aerosol composition such as the ammonium nitrate fraction, the acidity of the particles and the water content. A Nafion drier is installed in the sampling line to eliminate CE variations from water content fluctuations. Based on Fröhlich et al (2015) and after comparing the total mass of $PM_1$ from a Mobility Particle Size Spectrometer (MPSS) with that of the ACSM plus the equivalent Black Carbon (*eBC*) (Figure S2), the CE for this campaign was chosen to be 0.28. The resulting comparison between ACSM-derived sulphate with that from offline filters also provided consistent results (not shown).

The *eBC* concentrations were obtained from the absorption at 660 nm from the harmonized dataset of an AE31 aethalometer (Magee Sci.) and a Continuous Light Absorption Photometer (CLAP, NOAA), which sampled through a $PM_{10}$ cut-off inlet. The concentration of the light-absorbing aerosol is generally calculated from the rate of change of the optical attenuation of light on a quartz filter at seven different wavelengths (370, 470, 520, 590, 660, 880 and 950 nm) after correcting for loading and multiscattering effects (Backman et al., 2017, Stathopoulos et al., 2021). The number concentration at different size bins was determined using an MPSS (Rose et al., 2021). The PICARRO analyzer was deployed for measurements of the Greenhouse Gases (GHGs) $CO_2$, CO and $CH_4$. A Particulate Volume Monitor (PVM-100) (GERBER SCIENTIFIC INC., Reston, VA 20190, USA) (Gerber et al., 1999) was permanently installed at the station, which measures the Liquid Water Content (LWC) and the effective droplet radius of clouds by directing a diode-emitted laser beam along a 40 cm path with 1-hour time resolution. Meteorological data were obtained from a weather station installed at (HAC)[2]. Two high-volume samplers provided Total Suspended Particles (TSP) and $PM_{2.5}$ on filters that were afterwards analyzed by a SUNSET EC-OC analyzer (Diapouli et al., 2017) and XRF (X-Ray Fluorescence) spectrometer (Manousakas et al., 2018). Moreover, offline AMS analysis was performed on the TSP filters following the procedure of Vasilakopoulou et al. (2023) using a High-Resolution Time-of-Flight Aerosol Mass Spectrometer (HR-ToF-AMS), a state-of-the-art instrument that can provide continuous measurements of the atmospheric aerosol size distribution, concentration and chemical composition (Jayne et al., 2010; Drewnick et al., 2005). A pulsed Doppler scanning lidar system (StreamLine Wind Pro model, HALO Photonics) (Newsom et al., 2022) emitting at 1.565 μm was deployed at the VL site to estimate the PBLH, based on the standard deviation of the vertical velocity, combined with aerosol chemical composition metrics and humidity levels (Foskinis et al., in review).



## 3 Methods

### 3.1 Metrics for PBL influence at (HAC)[2]

Numerous methods are generally used to estimate the PBLH, including in-situ observations, remote sensing techniques and modelling based on meteorological data. The segregation between PBL-influenced and FT air masses is a challenging issue,
and given that there is no specific method that applies at all high-altitude sites, the local topography as well as the type of data available can generally determine the suitable methods for resolving PBL and FT air mass influence at a specific point. Both in-situ observations and modelling techniques have been used for this purpose. The most common approach is radiosonde measurements of temperature, humidity and/or wind profiles, although they lack in spatial and temporal resolution (de Arruda Moreira et al., 2018). The in-situ approaches include measurements of the water vapor mixing ratio (McClure et al., 2016,
Zhou et al., 2018), Radon-222 (Fröhlich et al., 2015, Farah et al., 2021), the $NO_y/CO$ ratio (Fröhlich et al., 2015), the relative increase in specific humidity between a low and a high-altitude station (Prévôt et al., 2000, Rinaldi et al., 2015), some statistical methods such as adaptive selection of diurnal minimum variation for $CO_2$: Yuan et al., 2018, or *eBC*: Sun et al., 2021) and remote sensing techniques (Doppler lidar, Aerosol Depolarization lidar). Trajectory models are also used to determine the boundary layer trajectories; FLEXTRA, based on data from the European Centre for Medium-Range Weather Forecasts
(ECMWF) and HYSPLIT are two common models used to retrieve the PBLH from meteorological data.

For the same campaign, Foskinis et al. (in review) retrieved the PBLH by the vertical profiles of the updrafts ($\sigma_w$) from the HALO Doppler Lidar installed at the VL site and linked the type of atmospheric layers to in-situ aerosol observations made on an hourly basis at (HAC)[2]. However, this dataset does not cover the whole period with ACSM data for the present study. We therefore examined a number of adjusted metrics to indicate the atmospheric layer, employing in-situ data and evaluated
their performance, while using the PBLH retrieved by HALO as a reference. The selected metrics included the water vapor mixing ratio (water vapor mass divided by the mass of dry air at a given air volume), the *eBC* to CO ratio and the accumulation mode number concentration (particles with diameter higher than 95 nm). Figure 1 shows the PBLH retrieved by HALO with respect to each metric: *eBC*/CO (a), Water vapor (b), and Accumulation mode number concentration (c).

FT air is generally very dry and PBL is generally contains about 80 % of the water in the atmospheric column (Myhre et al.,
2013), therefore the water vapor mixing ratio is considered an accurate indicator of PBL influence (Henne et al., 2005), especially in regions without considerable convective activity. The ratio of equivalent BC (*eBC*) to CO is a suitable proxy for determining fresh pollution arriving at (HAC)[2] from inside the PBL, in place of the $NO_y$ to CO ratio (Fröhlich et al., 2015, Farah et al., 2021) owing to a lack of $NO_y$ data at (HAC)[2] station. , CO is a gas emitted during incomplete combustion with a lifetime of several months in the atmosphere and is slowly degrading by OH radicals (Worden et al., 2013). *eBC* has a lifetime
of a few days. Their ratio in the FT is markedly different than the one in the PBL. Moreover, 90 nm is the average dry diameter threshold above which particles are activated to cloud droplets (Herrmann et al., 2015). Therefore, the number concentration of the particles in the accumulation mode (>95 nm) was another indicator for FT air masses.



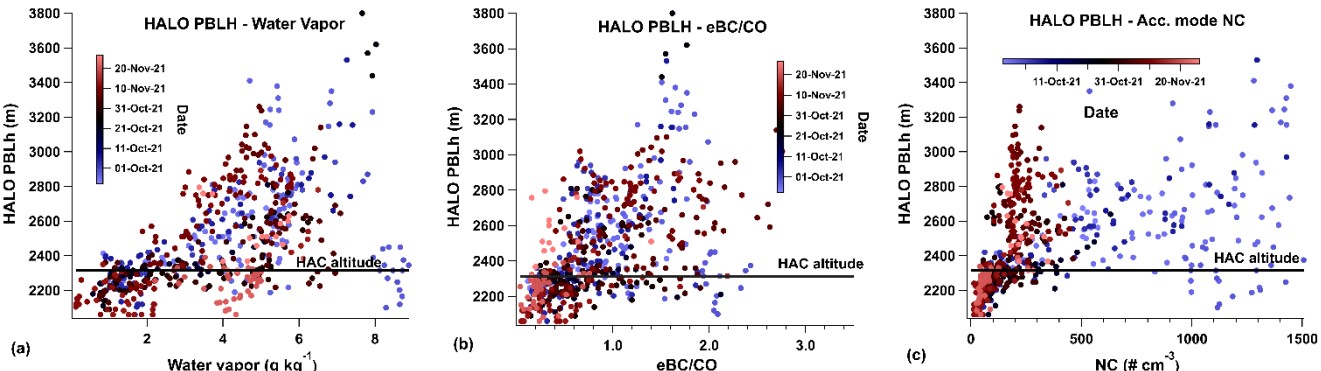

**Figure 1. PBLH as retrieved by the HALO Doppler lidar vs (a) water vapor ratio as a function of date, (b) *eBC*/CO ratio as a function**
**of water vapor ratio and (c) Number concentration of accumulation mode particles as a function of water vapor ratio.**

The PBL/FT thresholds for each method were chosen as the values maximizing the agreement between the metrics and HALO at FT conditions during the overlapping period. A threshold of 3.2 g kg$^{-1}$ was set on the water vapor mixing ratio based on the average of 5 years of measurements at (HAC)$^2$ (2017-2021) including only the winter months (December-February) when minimum influence from the PBL is expected (Zhou et al., 2018). The ratio of *eBC* to CO was used with a threshold of 0.5, while the number concentration of the accumulation mode particles (NC) threshold was chosen to be 100 cm$^{-3}$ in agreement with Gao et al. (in review) for (HAC)$^2$. By applying as criterion the 2 out of 3 metrics meeting the thresholds for FT conditions, an overall 85 % agreement was achieved and this combination was chosen for the segregation of FT from PBL conditions when remote sensing data were not available.

**3.2 Positive Matrix Factorization**

The Positive Matrix Factorization (PMF) technique was chosen to assign the NRS (both organic and inorganic ions) measured by the ToF-ACSM to different sources. PMF was performed on the combined ToF-ACSM dataset using the Source Finder Pro tool (SoFi Pro, Datalystica Ltd, Villigen, Switzerland) (Canonaco et al., 2021) that utilizes the multilinear engine ME–2 (Paatero 1999) as a PMF solver. The PMF model aims to describe the initial matrix **X** that contains information on the concentration of each variable in time as a product of the matrices **G** and **F**, where **G** is the source emission factor contribution and **F** is the spectral "fingerprint" (spectrum) associated with each factor. A residual matrix **E** is inevitably generated. The PMF principle is captured in Eq. (1):

$$\mathbf{X} = \mathbf{G}\,\mathbf{F} + \mathbf{E} \tag{1}$$

PMF aims to find the minimum of the quantity $Q$ ($Q_m$), which is the sum of the square of the ratio $e / \sigma$, as shown in Eq. (2):

$$Q_m = \sum_{i=1}^{m} \sum_{j=1}^{n} \left(\frac{e_{ij}}{\sigma_{ij}}\right)^2 \tag{2}$$

where $e$ is the residual and $\sigma$ is the uncertainty of each data point, $m$ is the number of rows of **F** and $n$ is the number of columns of **G**. This ensures that data with low signal to noise ratio (S/N) will be discarded so as not to affect the result.



The issue of rotational ambiguity, which makes it difficult for the model to arrive at an optimal solution due to the fact that **X** can be described with many different combinations of **G** and **F**, can be solved by applying certain constraints on **G** or **F** through the use of a-values. To assess the uncertainty of the solution, iterations with different starting points were performed using the bootstrap technique, which is described in more detail in Efron (2000).

At first, only the organic fraction of the ToF-ACSM was run in SoFi to identify the sources of organic aerosol at (HAC)[2], since no previous information is available for this high-altitude site. The solution that best fit the data consisted of three factors; one related to Primary Organic Aerosol (POA), and two oxidized secondary Organic Aerosol (OA); one more and one less oxidized (MO-OOA and LO-OOA, respectively). The same solution was reached by applying the PMF model on the offline dataset measured by filters from the high-volume sampler during this campaign and analyzed offline with an HR-ToF-AMS. Figure S3 shows the absolute concentrations of each factor from both analyses for each common day. The absolute off-line AMS concentrations were estimated from the percent contribution of each factor from the off-line analysis and the OA concentration reported by the ACSM. The MO-OOA factor contributed 50% on average to the OA, in good agreement with the 54% estimated by the on-line ACSM analysis for the same days. The off-line results confirm the presence of primary OA contributing 32% of OA. This value is a little higher than the 24% contribution estimated by the analysis of the ACSM data, however this difference can be explained by the uncertainty of the corresponding analysis. Finally, the LO-OOA contributed 14% to the OA according to the off-line analysis. These results provide additional support for the ACSM results and also demonstrate that the off-line method can provide useful information for the average source contributions in an area. The predicted day-to-day variation of the source contributions by the two methods differs more than the averages (Figure S3). Vasilakopoulou et al. (2022) showed that a significant part of this discrepancy is due to the low temporal resolution of the off-line AMS analysis. The rest is due to experimental issues (e.g. different water solubilities of the various OA components etc.). The details of the off-line analysis can be found in the Section S1 of the Supplement.

Subsequently, fully unconstrained simulations were performed on the combined ACSM dataset. The procedure for deconvoluting NRS sources was previously described in Zografou et al. (2022). In short, the variables of the inorganics that are characteristic for each species were added to the organics matrix, including $m/z$ 18, 32, 48, 64, 80, 81 and 98 of $SO_4^{2-}$, $m/z$ 30 and 46 of $NO_3^-$, $m/z$ 16 and 17 of $NH_4^+$ and $m/z$ 35 and 36 for $Cl^-$. The inorganic variables were downweighed before PMF analysis by a factor of $N^{1/2}$ (Ulbrich et al., 2009), where $N$ is the number of ions of each species that are duplicate according to the fragmentation table (Allan et al., 2004). The RIEs were applied beforehand separately for each species, followed by application of the CE.

The PMF analysis yielded five factors of the $PM_1$ fraction in Helmos station during the CALISHTO campaign; a primary organic factor (POA), ammonium nitrate (AmNi), ammonium sulphate (AmSul), one less oxidized OA (LOA) and one more oxidized factor (MOA) (Figure S4). The profiles of all the factors were extracted after unconstrained runs took place and were used as seed profiles for the next simulations. Five-factor simulations were then performed by constraining three of them (POA, AmNi and AmSul) and allowing for a variability of 30 % from the anchor mass spectrum (random a-values of 0.3) for 100 simulations, where the bootstrap technique was also enabled. This a-value was selected as the value that resulted in



minimum shift in the factors (Zografou et al., 2022). The POA and the secondary OAs presented extremely high correlation with the respective organic factors described before (Time series correlation, $R^2$ = 0.9-0.97). The POA consisted of 95 % organic ions, while MOA and LOA consisted of 80 % and 67 % organics, respectively. The MOA was mixed with 10 % $SO_4$ and 9 % $NH_4$ ions, while the LOA was mixed with 23 % $SO_4$ and 7 % $NH_4$ ions.

In order to evaluate the solution obtained, the residuals of the solution as well as the errors of each factor need to be addressed. The errors reported below are expressed as the spread of the factors to their median concentrations and are measured as the ratio of the interquartile difference ($75^{th} - 25^{th}$) to the median concentration; overall low uncertainties were found. AmSul displayed the lower variability at 1 %, then the secondary organic factors (both MOA and LOA) showed similar variability, at 3.7 % and 3.8 %, respectively. The POA's mass error was at 4.7 % and that of AmNi at 2.3 %. The probability density function

of the scaled residuals is shown in Fig. S5, in which it can be seen that most of the data fall into the suggested range of ± 3 % (Paatero and Hopke, 2003).

### 3.3 Analysis of back trajectories

Wind backward trajectories were obtained from the NOAA Air Sources Laboratory (ARL) Hybrid Single–Particle Lagrangian Intergrated Trajectory (HYSPLIT–4) model (Draxler and Hess, 1998; Stein et al., 2015). The 120h back trajectories were

calculated using the Global Data Assimilation System (GDAS) meteorological dataset at 1º resolution for every hour at the $(HAC)^2$ altitude. The Flexible Particle Dispersion Model (FLEXPART) was also used in order to obtain information on the geographical origin of the air masses at $(HAC)^2$ station, through the residence times of air parcels over geographic grid cells (Stohl et al., 2005). More details can be found in Vratolis et al (2023).

### 4 Results and Discussion

**4.1 PM₁ characterization and Source apportionment**

#### 4.1.1 Aerosol chemical characterization during CALISHTO

Chemical composition and concentration of $PM_1$ species found in the FT or at the interface between FT and PBL are expected to vary depending on different prevailing conditions, such as cloud formation, influence from PBL emissions and air mass origin. To account for this, a comprehensive characterization of $PM_1$ at $(HAC)^2$ during the CALISHTO campaign was initially

conducted and then the effects of clouds, PBL height and air mass origin were examined separately.

Figure S6 depicts the concentration of organics (green), sulphate (red), nitrate (blue), ammonium (yellow), and *eBC* (grey) in time with pie charts representing the fractions of each species at each month. Chloride was not included in this analysis, since it was close to or lower than the limit of detection for most of the campaign. In table S1 the average mass concentrations of each species, together with their relative contribution to $PM_1$ appears. Considerable variability was observed during the course

of the campaign for the $PM_1$ mass concentration levels and chemical composition, while concentration levels declined with





time towards the end of the campaign. During September, which is a transitional month with characteristics similar to the summer months in Greece and is more often influenced by PBL intrusions, the aerosol loading was up to 5 times higher than in October and November. Organics were the predominant aerosol species type during September, whereas higher sulphate levels were observed during October and November.  In September, sulphate made up 29% of the total $PM_1$ mass. By October,

it increased to 41%, and in November, it reached 47%. This reflects varying conditions at the Helmos station during the autumn months. The relative contribution of organics was 52 % during September and dropped to 36 % in October and 28 % in November. The ammonium contribution was fairly constant, varying between 11 % in September to 14 % in November. Aerosol nitrate was a minor contributor to mass (3 %). Equivalent black carbon progressively increased throughout the fall, from 5 % in September to 7 % in October and 8 % in November.

Figure S7 exhibits the diurnal variation of these species for each campaign month separately. Time in all plots is UTC+2 hours. The ToF-ACSM was operating 86 % of the time during CALISHTO. In September, all species exhibited similar daily concentration trends, with an increase starting at midday. This pattern is consistent with the peak in the PBL height at midday, which leads to an enrichment of anthropogenic emissions in the lower FT. The shallower PBL during the early morning and nighttime results in a drop in concentration, as well as change in the chemical composition. During October and November,

the organics and nitrate displayed similar patterns, as did ammonium with sulphate. The concentration of organics and nitrate was rising at midday during October and November, but with a longer duration in October, and a narrower peak in November. The duration and magnitude of the midday maximum values in $PM_1$ concentration shows a gradual decline from September to November. This behavior can be explained by the gradual decline of PBL influence at the $(HAC)^2$ altitude. Ammonium and sulphate, on the other hand, exhibited a similar trend in October, while their concentration remained more stable throughout

the day in November – reflective of the long-range transport influences controlling their levels.

### 4.1.2. Total NRS Source Apportionment through PMF

The POA factor retrieved by PMF is considered to include aerosol mixed from different primary sources that have had enough time to get mixed before reaching the $(HAC)^2$ station. This factor appears mainly when the station is under the influence of PBL air masses as will be discussed below and when the winds favor its vertical transport and provides valuable insight on

how primary sources (although mixed) from anthropogenic pollution can reach the FT and transfer pollution to high-altitudes. In Table S2 the correlation of this factor with external tracers is documented showing that this factor is impossible to be related to one single source, such as traffic or biomass burning and is representative of primary emissions mixed upon elevation at the station's altitude. Figure S8 depicts the time series and diurnal trends, as well as the mass fraction of each PMF factor for each campaign month, while Table S3 shows the monthly absolute concentration and the relative abundance of each factor. With

an average mass concentration of 0.19 μg m$^{-3}$, the relative contribution of POA at the end of the campaign drops at 7 % compared to 16 % at the beginning of the campaign. AmNi presented the lowest relative contribution to total $PM_1$, contributing 3-4 %. In the span of the campaign, this factor decreased 7 times from 0.14 μg m$^{-3}$ in September to 0.02 μg m$^{-3}$ in November displaying the character of a short-lived species. The AmSul concentration shows the least variability during the course of the





campaign regardless of the PBLH with respect to the station altitude due to its origin from long range transport and long

lifetime. This factor is the main contributor during the campaign representing 33 % of $PM_1$ in September, while in October and November its contribution increases to 53 and 60 % respectively. Concerning MOA and LOA, their mass decreased over the course of the campaign, from 46 % relative abundance in total in September to 30 % in November. The diurnal pattern of the PMF factors during the three months is influenced by aerosol originating within the PBL, as will also be discussed in Section 4.3.

To elucidate the dominant mechanisms leading to changes in concentration a number of processes need to be examined. Firstly, cloud processing can significantly impact the aerosol concentration by activation of particles into cloud droplets, as well as processes such as in-cloud scavenging. A second factor is the PBLH in relation to the station altitude, which determines whether the station was influenced by aerosol originating from within the PBL or lied in the free troposphere, where low concentrations of background aerosol are found. Finally, it is important to consider the origin of air masses. The above effects

on the observed $PM_1$ aerosol composition are discussed in the following sections.

**4.2 Aerosol composition during in-cloud periods**

The cloud periods were determined by the LWC given by a PVM-100 cloud probe, which also provided the effective radius of the cloud droplets, together with the RH using a threshold of 97 % where cloud presence was presumed for higher values. The LWC of typical clouds is in the range of 0.1 to 3 g m$^{-3}$; hence, this threshold was used to determine in-cloud conditions

(Seinfeld and Pandis, 2006, Roth et al., 2016).

As a first proxy for the influence of cloud periods on aerosol mass concentration, in Fig. S9, the bar graphs represent the mass concentration of NRS and *eBC* (Fig. S9a) and the PMF factors (Fig. S9b) for both in-cloud (referred to as IN-C) and no-cloud conditions (referred to as OUT), along with the relative abundance of each species and each factor, respectively. Significant differences are observed both in concentration levels and in chemical composition. Table S4 contains the respective average

concentrations in these two conditions for each species and each factor. Under clear sky conditions the organics present a more important contributor to $PM_1$, while under cloud conditions $SO_4$ is more important. In the same way, the factor AmSul is more important than the organic PMF factors (44 % as a sum) in-cloud, while under no cloud conditions the organic factors are dominant (59 % as a sum over 38 % AmSul). Compared to the organics, whose concentration is 3 times lower during cloud periods than in no-cloud conditions (similarly to those of ammonium), sulphate removal, due to collision of particles with

existing cloud droplets (in-cloud scavenging) and/or activation to cloud droplets, is less effective, presenting 2 times lower concentrations during cloud periods. However, this could also be related to simultaneous production of $SO_4$ due to $SO_2$ oxidation under aqueous conditions in clouds. The *eBC* shows similar decrease in-cloud as the organics, with 3 times lower concentration compared to no-cloud periods, while $NH_4$ and $NO_3$ are 2 times lower in clouds.

Although Fig. S9 gives a general picture of cloud influence on the chemical composition of $PM_1$, more specific details are

provided by Fig. S10, which presents the mass concentration of each species and factor for 3 conditions: pre-cloud aerosol (1 hour before cloud formation), during cloud presence and post-cloud aerosol (1 hour after cloud). The respective graphs for the



PMF factors appear in Fig. S11. The in-cloud scavenging of aerosol is found to cause a reduction in their concentration by a factor of 2.5-3. Organics and sulphate show the highest decrease during in-cloud periods, due to the high hygroscopicity and scavenging of some organics and sulphate. In contrast, these two species are found to display a concentration increase in post-

cloud time periods, possibly related to $SO_4$ and SOA production from aqueous oxidation of $SO_2$ and VOCs, respectively, which is consistent with Ervens et al., (2018). LOA is the PMF factor with the highest increase post-cloud. It is notable that AmNi shows negligible increase after cloud processing, which is related to negligible uptake of $N_2O_5$ and $NO_3$ in clouds (Hauglustaine et al., 2014) that would result in AmNi formation in presence of ammonia. It has to be noted though that the the in-cloud instances include both interstitial and activated aerosol.

In order to separate between activated droplets sampled and interstitial (particles that remained non-activated during the in-cloud periods, either due to unavailability of LWC or due to size limitations) a "virtual filtering technique" (Foskinis et al., in review) took place. Studying the average particle distribution during pre-/post- and in-cloud conditions, Foskinis et al (in review) exploited the $PM_{10}$ sampling lines at (HAC)[2] used for the aerosol in-situ measurements and found that cloud droplets with diameter less than an empirically observed by the PVM-100 threshold of ambient effective droplet diameter ($D_{eff}$) at 13.5

μm were susceptible to enter the sampling line, get dried and return to the actual size before activation and therefore be detected as part of the measured number size distribution. A general rule was followed according to which when the $D_{eff}$ was lower than 13.5 μm, the aerosol measured was considered to contain both activated and interstitial aerosol, while at certain periods when the $D_{eff}$ was higher than 13.5 μm only interstitial aerosol could be measured. Considering this, an approximation of the activated fraction could be estimated as the difference between cloud free aerosol (1 hour before cloud formation) and

interstitial aerosol. To this end, in Fig. 2 the box plots of the free and interstitial aerosol are plotted, together with the average value as approximated for the activated part for the NRS and *eBC*. The respective graphs for the PMF factors appear in Fig. S12. The most efficiently activated specie is $SO_4$ with 84 % activation rate, which is reasonable considering that sulphate is a highly hygroscopic component. The lowest activation rate appears for $NH_4$ at 67 %. This difference is explained by the different size distribution of activated particles. Foskinis et al. (in review) showed that activated particles present a shift in the size

distribution towards higher diameters, where ammonium and sulphate are mainly in the ammonium bisulphate form rather than in the ammonium sulphate form which is more dominant in lower size distributions (Mészáros and Vissy, 1974). In addition, entrainment of FT air, which is more acidic, can also explain this behaviour. Looking at the respective plots for the PMF factors, those with highest activation rates are AmSul and MOA.





**Figure 2. Organics (a), SO₄ (b), NH₄ (c), NO₃ (d) and *eBC* (e) box plots for cloud-free (1 hour before cloud formation) (Free), interstitial (Int) and activated aerosol (Act) (only the average value as the difference between the mass concentration before cloud formation minus the mass concentration of the interstitial part of the aerosol). The boxes range are the 25th and 75th percentiles, while the whiskers ranges are the ±Standard Deviation. The median is described as a horizontal line, while the rectangular represents the average value.**

In order to eliminate the influence of cloud events on the subsequent analysis, the results presented in Sections 4.3 and 4.4 refer to non-cloud periods only.

## 4.3 PBL influence on chemical composition

In Fig. 3 the concentration box plots of the concentration of NRS and *eBC* (Fig. 3a) and the PMF factors (Fig. 3b), together with their relative abundance, appear segregated between PBL and FT conditions for the whole campaign. The respective mean values appear in Table S6. There are great differences observed in the loadings of $PM_1$ in PBL and FT conditions. The total $PM_1$ concentration reaches an average value of 2.8 µg m⁻³ when the station is influenced by PBL, while it drops to only 0.5 µg m⁻³ under FT conditions. Nitrate yielded the highest ratio of PBL/FT concentrations close to 8. The measured $NO_3$ by ACSM is particulate nitrate formed by the conversion of $NO_x$ to the particle phase. $NO_x$ is quickly depleted; therefore, nitrate is only formed inside the PBL resulting in much lower concentration in the FT, arising from injections from the PBL. Organics and $SO_4$ followed with a ratio close to 6, then $NH_4$ with PBL/FT equal to 4, while finally the lowest ratio (PBL/FT=3) was observed for *eBC*.

Organics are the dominant species in both conditions (which is in agreement with Zhou et al., 2018), followed by $SO_4$. $NO_3$ contributes the same in both conditions, $NH_4$ shows higher relative abundance in the FT than in the PBL (16 % over 11 %), while *eBC* is twice as high in relative terms in the FT than in the PBL (10 % of $PM_1$ in the FT over 5 % in the PBL).

Overall, during the CALISHTO campaign the NRS composition (that is excluding *eBC*) did not change much, as is evident also by the very similar composition of the PMF factors in both PBL and FT conditions. It is possible that a difference in the composition would be observed in winter time when the station would stay for longer times in the FT, and possibly higher sulphate relative abundance would be observed. During autumn, there are no clean periods, where the station stays at the FT for several days; there is repeated injection of PBL pollution in the FT. This is interrupted by some continuous FT periods, which however last less than the lifetime of the species introduced. Therefore, the chemical composition does not vary significantly inside and outside the PBL, although the PM loading does vary depending on whether there is exposure to PBL air masses. Thus, it can be seen that the factors contribute equally in both conditions regardless of whether they originate from long-range transport, like AmSul, or have longer lifetimes, like MOA. Nevertheless, the increased relative persistence of *eBC* levels at higher ratios than other species in the FT when compared to PBL levels can pose serious climatic implications, since the direct radiative forcing caused by the *eBC* is more important than that of other species. This is consistent with the study by Zhang et al., 2017, where this increase was related to Brown Carbon absorbance. This deserves further study for carbonaceous aerosol at (HAC)[2].







**Figure 3. Box plots of NRS and eBC (a) and PMF factors (b) for PBL and FT conditions separating based on the criterion of 2 out of 3 methods. The boxes range are the 25th and 75th percentiles, while the whiskers ranges are the ±Standard Deviation. The median is described as a horizontal line, while the rectangular represents the average value.**

Figure 4 depicts the diurnal trend of the mass concentration of the NRS and *eBC* separated by whether the station was inside the PBL or in the FT, together with the diurnal PBLH variation from HALO, while in Fig. S13 the same plots appear for each PMF factor. It is obvious that all species, as well as the PMF factors, in PBL conditions follow the same diurnal trend as the PBLH, except for $SO_4$ and AmSul, which is expected, since AmSul is mainly long-range transported aerosol, and therefore



not that sensitive in daily fluctuations of the PBLH and moreover SO₄ can also be produced in the FT. The midday peak is
observed between 10:00 AM and 18:00 AM (at UTC+2 time zone), driven by greater convection related to increased solar
radiation at this time. In the FT small fluctuations are observed, that are rather random and do not follow a standard pattern
between the species or factors.

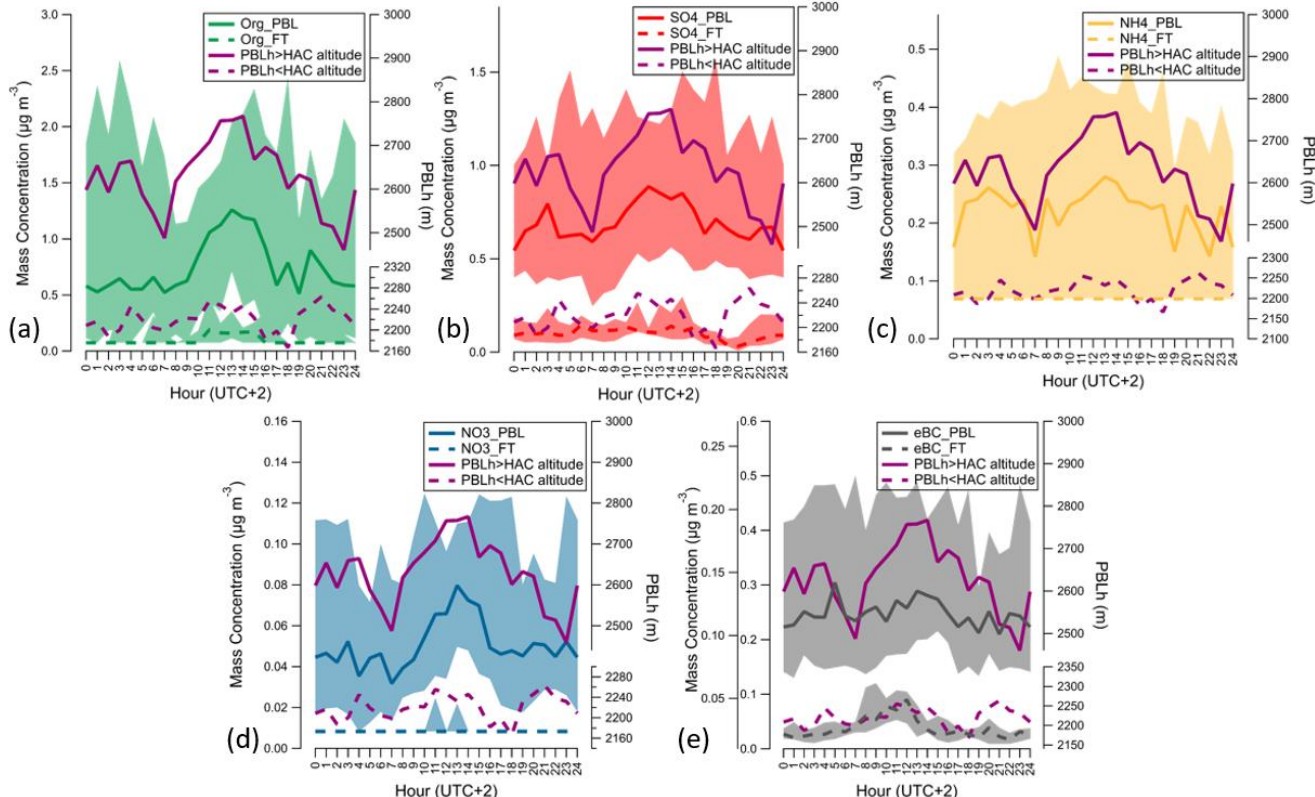


**Figure 4. Median and interquartile (10th and 90th) diurnal trends for each NRS species (a: Organics, b: SO4, c: NH4, and d: NO3)
and *eBC* (e) for the whole campaign segregated between PBL-influenced days and days in the FT based on the criterion of at least 2
methods.**

**4.4 Air mass origin influence on chemical composition**

The (HAC)² station lies in a crossroad of different incoming air masses and the aerosol presents different characteristics
depending on the incoming origin. The back trajectories analysis allows the differentiation of the air masses arriving at the
(HAC)² into four different categories as appear indicatively in Fig. S13: Dust (D) when the air masses arrived from North
Africa, Continental (C) arriving from Europe and mainly from Western Europe, Marine (M) either from the Mediterranean or
the Adriatic Sea and the combination of Marine and Dust (M-D). The difference between D and M-D is that D back trajectories
show higher residence times over North Africa, while M-D show equally shared residence times over North Africa and either
Adriatic or Mediterranean Sea. In Fig. 5 the bar graphs show the mass concentration of PM₁ species (Fig. 5a) and the PMF



factors (Fig. 5b), together with the respective percentage that represents their relative abundance, for each of the previously described origins.

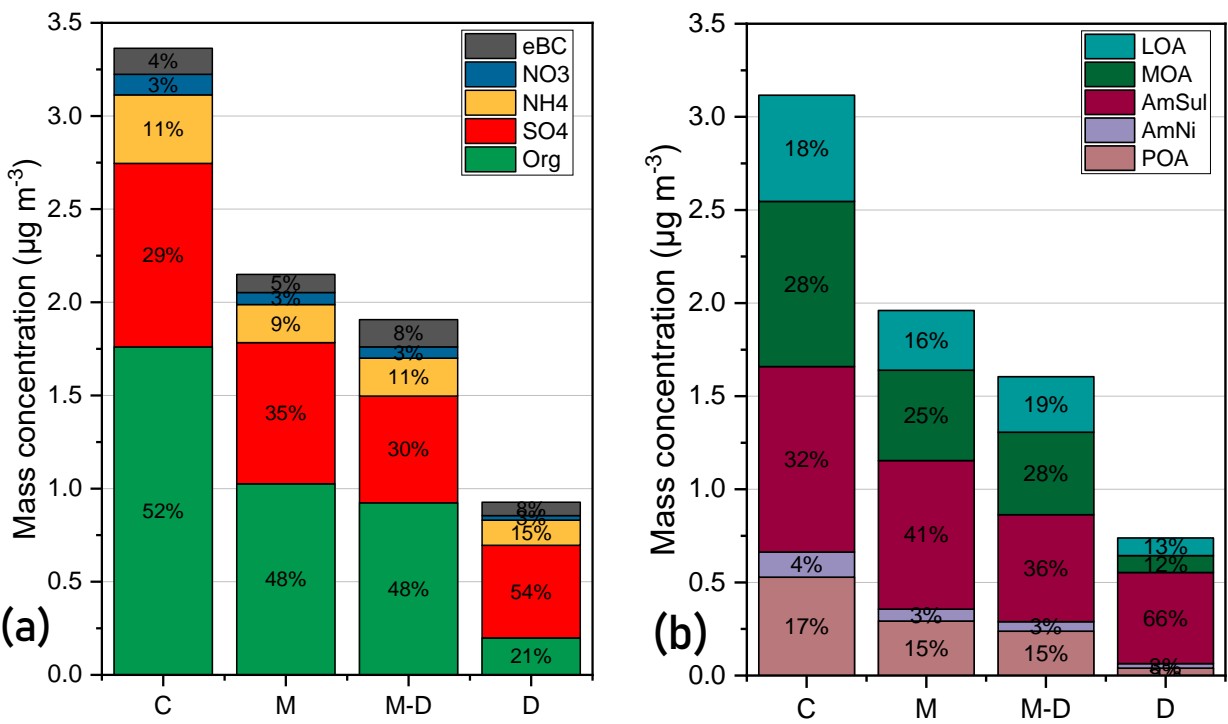

**Figure 5. Bar graphs representing the mass concentration of the PM₁ species and PM₁ factors, where C: continental, D: Saharan Dust, M: Marine aerosol and M-D: Marine and Dust.**

In general, great differences were observed between Dust air mass origin and the other aerosol types both in terms of PM loading and aerosol composition. The aerosol with highest mass loading was Continental with 3.4 µg m$^{-3}$. Marine and Marine-Dust followed with total PM₁ equal to 2.2 µg m$^{-3}$ and 1.9 µg m$^{-3}$ respectively, and then Dust events with 0.9 µg m$^{-3}$. This in accordance with Carbone et al (2014), which found the North African air mass origin at Mount Cimone (2165 m a.s.l.) to carry the less PM mass loading. An interesting finding is that Dust events are related with the highest sulphate fraction and the lowest organics fraction, and together with Marine-Dust aerosol exhibit the highest relative abundance of *eBC* at 8 %. In absolute terms, *eBC* is higher in Continental aerosol, which is known to carry important amounts of pollution from industrial and biomass burning plumes, and in this case is richer in organics than all the other origins and shows the lowest AmSul relative abundance. M-D is shown to be mainly affected by Marine aerosol with closest concentration levels and more similar composition. Ammonium sulphate is greatly important in aerosol originating from North Africa with 66 % relative abundance. This is probably related to aerosol passing through the Mediterranean while being transferred from North Africa, getting enriched with non-sea-salt sulphate, which commonly appears in marine environments, while transformation processes during transport result in ammonium sulphate formation. Consistently, Marine and Marine-Dust aerosol also carries an important



fraction of Ammonium Sulphate (41 % and 36 %, respectively). MOA is again the prevalent OA factor in all aerosol types, followed by POA and LOA. AmNi is not seen to be affected by air mass origin.

## 5. Conclusions

This is the first study presenting results on the chemical characteristics of $PM_1$ aerosols at the $(HAC)^2$ station, the only high-altitude station in the Mediterranean, where measurements and analysis of this kind have been conducted during the
CALISHTO campaign. The PMF analysis apportioned the $PM_1$ mass as follows: two secondary inorganic aerosol components, ammonium nitrate (AmNi) and ammonium sulphate (AmSul), one primary (POA) and two secondary organic aerosol components: one more oxidized (MOA) and one less oxidized aerosol (LOA). The results of the OA PMF were also supported by the results of PMF after offline AMS analysis on filter extracts took place. The POA factor identified here was linked to a mixture of primary sources that arrive at the station before undergoing oxidization, but could not be attributed to a single
primary emission source. The $PM_1$ characterization was carried out using 3 classifications of air masses sampled: in-cloud/cloud-free, interstitial PBL/FT conditions, and air mass type. Cloud presence resulted in lower $PM_1$ concentrations due to particle activation and cloud scavenging. Sulphate, although dominant in both in and out of cloud conditions, is more influenced by clouds than organic species (greater concentration decrease). $SO_4$ and organics were found to replenish faster their concentrations after cloud events compared to the other species (ammonium, nitrate and $eBC$), pointing to $SO_4$ and
organics formation in-cloud following aqueous-phase oxidation of $SO_2$ and VOCs, respectively. The separation of interstitial and activated particles during cloud events led to the conclusion that interstitial aerosol is richer in low hygroscopicity organics and more acidic inorganics. Some metrics were evaluated as to their ability to identify FT over PBL conditions at $(HAC)^2$ station, taking as reference the PBLH from parallel measurements by a HALO Doppler wind lidar. PBL conditions, in comparison to FT, were related to much higher mass concentration of all species. Concerning aerosol origin, it was found that
air masses coming from Continental Europe (C) carried the highest levels of $PM_1$ pollution; twice as high as Marine (M) and Marine enriched with Dust (M-D), and thrice as high as Dust from North Africa (D). Sulphate was the most abundant species in Dust aerosol (and AmSul was therefore the most abundant PMF factor), indicating influence from marine non-sea salt $SO_4$ uptake during transport from the North Africa to $(HAC)^2$ passing through the Mediterranean Sea.

As an overview, it was found that cloud processing influences both aerosol loading and chemical composition. Aerosol
loadings within the PBL were 5 times higher on average compared to those in the FT, while the chemical composition or the source-apportioned components for the inorganic and organic fractions remained rather unchanged. An exception was the $eBC$ concentrations with a higher relative abundance in the FT. This is a key finding that needs to be studied further.




*Author contribution*

AN, AP and KE organized the CALISHTO campaign. OZ, KE and AN conceived and led this study. OZ led the data analysis,
and interpreted the results with contributions from KE, AN, MG, RF, SNP, and ED. OZ wrote the original manuscript with
inputs from KE, AN, RF, SNP, ED and CNV. OZ, KG, PF, MG, KE, RF, AP and AN conducted experiments and collected
the raw data. SNP, ED and CNV did the offline AMS analysis, and run the PMF for the AMS data. SV performed FLEXPART
simulations and OZ performed HYSPLIT simulations. All authors discussed, reviewed and edited the manuscript.

*Competing interests.*

The authors declare that they have no conflict of interest.

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
