# Peer review of "High Altitude Aerosol Chemical Characterization and Source Identification: Insights from the CALISHTO Campaign"

_EGUsphere, 2024_

## Author Comment (AC1)

**Reviewer 1**

The manuscript by Zografou et al. describes measurements of aerosol composition collected at a high-altitude site and discusses aerosol-cloud interactions and differences between the boundary layer and free tropospheric aerosol. The manuscript is clear and relevant to the scientific community. I recommend for publication with the following minor revisions.

> We thank the reviewer for the positive assessment! Responses to the points raised are provided below.

**Specific Comments:**

Section 2: I don't think the measurement duration is stated in the main text. Please include the dates (or at least number of days) in section 1 or 2.

**> Done**

Line 130: "During the CALISHTO campaign (September-November 2021),..."

Section 3.1: It would be helpful to include the proportion of measurements collected in the FT versus in the BL. Additionally, what portion of the time has the collocated HALO measurements that were used for verification, and what portion was separated utilizing the thresholds described in section 3.

> We thank the reviewer for the suggestion. The above information was added to our manuscript as follows:

Lines 208-210: "The separation of PBL and FT air masses was conducted using HALO for 30 % of the campaign, while for the remaining period, the set of metrics was utilized. Moreover, the proportion of FT to PBL air masses was 1:3."

I believe that more clarification is needed about the virtual filtering technique that is used. On line 360 it is stated that when Deff is < 13.5  $\mu$ m, it is assumed that the composition reflects both the interstitial and residual, and when Deff is > 13.5  $\mu$ m the composition reflects only the interstitial aerosol, however this still leaves me with some questions. Does this method account for the droplet size distribution at all. For example, if the Deff is slightly above the threshold, there are still likely to be some droplets smaller than 13.5 $\mu$ m. Will these still be sampled and categorized as interstitial?

> These are all good points and are extensively discussed in the Foskinis et al (2024b) paper. The "virtual filter" technique was developed to ensure that the aerosol sampled by the  $PM_{10}$  inlet does not contain aerosol from evaporated droplets, but only interstitial aerosol, based on the sensitivity analysis of the measured aerosol size distribution. Thus, the optimum  $D_{eff}$ , came up as the minimum value for which the measured aerosol size distribution became insensitive to the chosen threshold value. More information can be found in detail in the Foskinis et al (2024b) paper where Figure 3 displays an example of this process applied to a segment of data from CALISHTO. We refer to a summary of the main points in the text (lines 373-383).

The citation of the paper by Foskinis et al (2024b) that is now in ACPD was updated accordingly: Lines 571-573: "Foskinis, R., Motos, G., Gini, M. I., Zografou, O., Gao, K., Vratolis, S., Granakis, K., Vakkari, V., Violaki, K., Aktypis, A., Kaltsonoudis, C., Shi, Z., Komppula, M., Pandis, S. N., *Eleftheriadis, K., Papayannis, A., and Nenes, A.: Drivers of Droplet Formation in East Mediterranean Orographic Clouds, https://doi.org/10.5194/egusphere-2024-490, 25 March 2024.*"

A description of the statistics of the duration of the cloud events is needed to provide confidence in these comparisons. How many cloud periods occurred, and what fraction of these were filtered as interstitial or activated. I might recommend including a time series of Deff in supplemental material - for example in conjunction with Figure S6.

> Here, we understand the reviewer's concerns, but given that all these points referred to the backbone of the manuscript by Foskinis et al (2024b), we consider it wise not to include such a detail within the manuscript. The time series of  $D_{eff}$  are already included in Foskinis et al (2024b) Figure 2e.

What are the main variables effecting the Deff at this site. Is there a possibility that the events with Deff above the filtering threshold are systematically different (in regards to sources and/or aerosol composition) than those with lower Deff.

> This is also extensively discussed in the Foskinis et al. (2024b). Generally speaking, the aerosol variations (not vertical velocity variations) are driving the average droplet size, and these in turn are driven by whether the aerosol originates from the FT or the PBL. Droplets are smaller and clouds tend to be velocity limited when the aerosol originates from the PBL - while the droplets are larger and clouds tend to be aerosol-limited when the aerosol originates from the FT.

I am slightly wary of figures 2 and 3. In figure 2, why does the interstitial box plot have error bars for 1SD, but no 25-75% area? Likewise, why does the activated box have neither. In figure 3, the organic FT has no 25-75 box.

> This is because most values in the case of interstitial aerosol are close to the detection limit of each species, and therefore no statistics appear. As for the standard deviation (SD) error bars, it was calculated based on the available data points, even though they are limited by the detection limit.

The activated part, as discussed on the manuscript, is only an average value indicative of activated species, estimated as the difference of the mean value of mixed interstitial/activated aerosol minus the interstitial-only aerosol, and is therefore depicted only as a mean value.

Section 4.4: two of the backtrajectories contain the label "dust". Is there direct evidence that the aerosol actually contains dust particles (i.e. increase in TSP mass or evidence from the XRF measurements), or is this an assumption based on the back trajectory. As the focus of section 4.4 is on the submicron non-refractory aerosol composition, which is less likely to be directly impacted by dust, this nomenclature is slightly confusing as it seems probable that these periods are influenced by long range transport of pollutants. However, if there is evidence of these chemical differences being related to the presence of dust, this would be an important finding and should be more clearly stated.

> Absolutely. The first evidence was the route of the back trajectories originating from the Adriatic or the Mediterranean that were also passing from North Africa. And it was also

confirmed by cross-checking the data from a Nephelometer installed at the station (from the Ångström exponent of the scattering coefficient) and the SKIRON model. Moreover, the study of Gao et al. (2024, ACPD) shows that for this period, dust influence is clear (presence of large particles that are not fluorescent). We also refer to the relevant study for more information.

Technical corrections: Line 177: Missing parenthesis on citation > Done

**Line 332: The LWC threshold used for defining the cloud periods is not clearly stated.**

> The threshold was 0.1 g m-3 for cloud presence. It was rephrased in the manuscript to be clearer.

Line 341: "The LWC of typical clouds is in the range of 0.1 to 3 g m-3; hence, this a threshold of 0.1 g  $m^{-3}$  was used to determine in-cloud conditions (Seinfeld and Pandis, 2006, Roth et al., 2016)."

Line 360: Whether a droplet activated also depends on aerosol composition as non-hygroscopic aerosol (i.e. most POA, eBC, etc) will not activate event with sufficient LWC or adequate aerosol size.

> LWC depends on the history of cooling in the parcel, while droplet formation depends on the dynamical forcing (vertical velocity/cooling rate) and the amount of aerosol present that compete for supersaturation. We have included any clarifications necessary in the revision of the manuscript.

Lines 373-375: "In order to separate between activated droplets sampled and interstitial (particles that remained non-activated during the in-cloud periods, either due to chemical composition, small dry size or insufficient vertical velocity, that results in cloud maximum supersaturation below the critical value required),..."

**Reviewer 2**

The authors present a statistical analysis of chemical composition measured using a time of flight aerosol chemical speciation monitor at a mountain top site located in Greece. The analysis focuses on contrasting aerosol composition in the free troposphere and planetary boundary layer, and between cloud free, interstitial and activated aerosol. Positive matrix factorization is used to identify several factors contributing to the organic aerosol mass. I recommend the manuscript for publication after the following comments are addressed.

> We thank the reviewer for the positive assessment! Responses to the points raised are provided below.

It would be helpful to more effectively summarize if the mass fractions of the different aerosol species - including the OA factors - vary between the PBL and the activated aerosol. It is probably buried somewhere in Figure 2, but it's hard to quickly extract from the presentation. It wasn't clear from the text to me what an 84 % activation rate really means.

> Thank you for pointing this out. We now have clarified what the statistics mean. As activation rate here is referred the ratio of activated to cloud-free (30 min before cloud formation) mass concentration of each specie:

*Line 387: "The most efficiently activated specie is SO4 with 84 % activation rate (determined here as the ratio between activated and cloud-free mass concentration),..."*

It would seem to be important to better explain the in-cloud sampling technique, as it is new and not well characterized. Especially detailed discussion on how it differs from a traditional CVI inlet are critical to include. What is the precise cut size? How broad is the cut size? What are the transmission efficiencies of the transmitted cloud droplets? Is there an upper limit? Is there broad number closure (maybe even just with modeled CDNC since only the PVM was available) between CDNC passing through the inlet and measured residual aerosol number concentration? Perhaps these details are derived in the submitted papers, but it needs to be presented here as well.

> This point has also been raised by reviewer 1. We include a short summary of the technique (lines 373-383) and point to the related companion manuscript by Foskinis et al (2024b), where all the points raised (and more) are comprehensively discussed.

The structure of the text could be improved. There seemed to be significant discussion of results with reference to Figures in the supplementary information. It would seem to me that many of these results should be promoted to the main text and/or discussion in the text referring to supplementary figures should be de-emphasized.

> We understand this point, and have shifted somewhat some parts of the discussion from the supplement to the main text. However, we do not feel that extensive rearrangement of the manuscript (which is already long) is necessary.

We promoted Fig. S6 to the main text as Fig. 2 in Section 4.1.1, and also moved Fig. S9 in Section 4.2 as Fig. 3.

There are two Foskinis et al. papers in review. Please distinguish in the citations. > Thank you for pointing this out. We have now cited them as "a" and "b"